# Radar Spectra-language Model for Automotive Scene Parsing

## Abstract

Radar sensors are low cost, long-range, and weather-resilient and provide direct velocity measurements. Therefore, they are widely used for driver assistance functions, and are expected to be crucial for the success of autonomous driving in the future. In many perception tasks only pre-processed radar point clouds are considered. In contrast, radar spectra are a dense and raw form of radar measurements and contain more information than radar point clouds. However, radar spectra are rather difficult to interpret. In this work, we aim to explore the semantic information extracted from spectra in the context of automotive driving, thereby moving towards to a better interpretability of radar spectra. To this end, we create a radar spectra-language model, allowing us to query radar spectra measurements for the presence of scene elements by using free text. We overcome the scarcity of radar spectra data by matching the embedding space of an existing vision-language model (VLM). Recognizing that off-the-shelf VLMs underperform on our target domain, we develop a fine-tuning approach tailored to automotive scenes. Finally, we explore the benefit of the learned representation for scene parsing, obtaining improvements in free space space segmentation and object detection merely by injecting the spectra embedding into a baseline model.

## 1 Introduction

Radar has emerged as a valuable sensing modality in the automotive domain, combining the benefits of low hardware cost with long-range and weather-resilient sensing, as well as direct velocity measurements. These properties are highly important in autonomous driving applications. Therefore, radar sensors are already used for driver assistance functions and are also expected to play an important role in autonomous driving in the future. Perception progress has been limited by specifics of the sensing modality: Radar emits and measures the reflections of electromagnetic waves and it often struggles to detect objects that are small and less reflective (e. g., a pedestrian). This becomes even more difficult if a poor-reflecting object appears alongside stronger reflectors such as large metal objects (like a truck, which often appears in driving scenes).

Digital signal processing, involving usually multi-dimensional Fourier transformations, is used to convert the measured baseband time signal to a representation known as radar spectra. Spectra are represented as complex valued tensors with shape and dimension dependent on the radar sensor at hand. As an example, tensor axes correspond to range, Doppler velocity, and antenna channels, where the latter dimension corresponds to the data received by each antenna channel (Rx), cf. Fig. 1a. Another typical representation are a sequence of range-azimuth tensors, where each element in the sequence corresponds to one transmitted chirp of the radar sensor. Spectra are processed to suppress noise and filter out local intensity maxima, the radar reflections, resulting in a list of radar reflections called a radar point cloud. An overview illustration of the above signal processing chain is provided in Appendix A.1.

Numerous radar perception algorithms are based on radar point cloud data as input Ulrich et al. (2022); Danzer et al. (2019); Svenningsson et al. (2021). Nevertheless, in the point cloud computation, information that is available in the raw spectral radar data, is inevitably lost. Recent work (Major et al. (2019); Cozma et al. (2021); Wang et al. (2021); Paek et al. (2022); Rebut et al. (2022); Zhang et al. (2021)) shows that perception algorithms applied on radar spectra can achieve improved performance. Spectra might be especially useful for objects which are hard to detect, like pedestrians, which usually result in only few radar reflections. Nevertheless, working on radar spectra

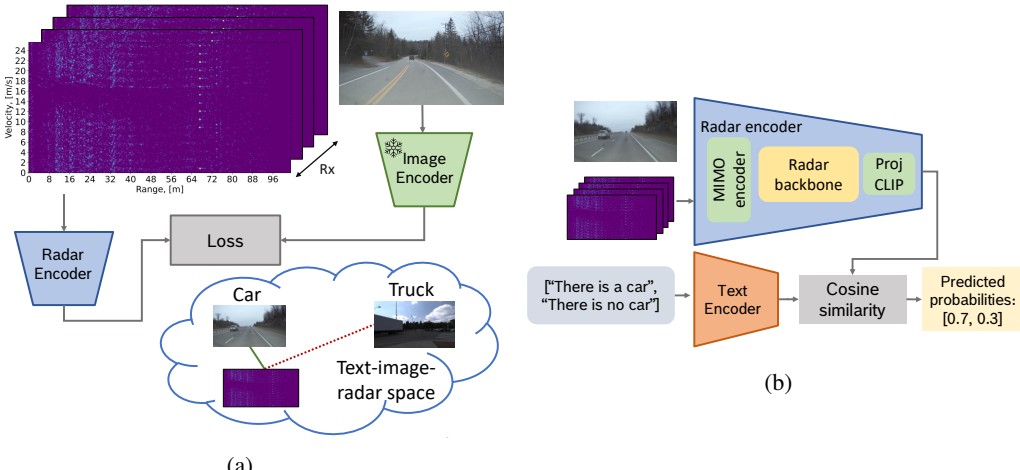

(a)

(b)

Figure 1: Training of a radar spectra-language model utilizes a frozen image encoder taken from a vision-language model for supervision. (a) Radar spectra encoder is trained to match embeddings of the corresponding RGB images. (b) Radar spectra-language model inference.

introduces new challenges: First of all, there are only a small number of labeled datasets available which include radar spectra. Furthermore, the spectra data is difficult to interpret by humans, as evidenced by Fig. 1a depicting the RGB image alongside its corresponding range-Doppler spectrum. This leads naturally to the question: what scene information is captured in radar spectra?

To address the above, we propose to train a radar spectra-language model (RSLM) for automotive scenarios, motivated by the tremendous success of vision-language models (VLMs) like CLIP (Radford et al. (2021)), or DALL-E (Ramesh et al. (2021)). Said model can subsequently be used to query radar measurements for contents of interest using free text, a step towards understanding the semantic content of radar spectra, alleviating the difficulty of their interpretation by humans.

To train the radar encoder, we utilize the frozen image encoder of a VLM (see Fig. 1a). During training, the radar encoder embeddings are forced to match the feature embeddings of the image encoder. In the VLM the output feature embeddings of the text encoder are aligned to the feature embeddings of the image encoder, i. e. text domain is connected to image domain. Since we align the feature embeddings of the radar encoder to the ones of the image encoder, the feature embeddings of the radar encoder are aligned to the ones of the text encoder as well, i. e. text domain is connected to radar spectra domain (see Fig. 1a). In this way, we construct the *radar spectra-language model*. To the best of our knowledge, we are the first to train a radar-language model. Note that for training the radar encoder only paired image-radar spectra samples are needed, no labeled spectral radar data is necessary. This tackles the problem of a large labeled radar spectra dataset, which is usually needed for a supervised training.

Here we are especially interested in automotive applications, and therefore benchmark different VLM for classification in automated driving scenes with the aim of selecting a reference model for training a radar encoder for the radar spectra-language model. Since the performance is not satisfactory, we propose a method to fine-tune the VLMs for automotive scenes. To explore the semantic content of radar spectra, we benchmark the RSLM on scene retrieval tasks: Free text is used to describe a scene, and the RSLM is used to search for data samples which fit to this scene description. Moreover, to investigate the benefits of the semantic prior learned by the RSLM, we combine the trained radar encoder with an object detection and segmentation network. The performance on two downstream tasks, object detection on radar spectra and free space space estimation, is evaluated.

Our main contributions can be summarized as follows:

- We propose training and evaluation of the first radar spectra-language model.
- We benchmark scene retrieval using the radar spectra-language model, exploring semantic content of radar spectra.
- We investigate the benefits of the learned radar feature embeddings on two downstream tasks: object detection and free space estimation.

- We conduct a benchmark comparison of off-the-shelf vision-language models (VLMs) for classification in automotive scenes.
- We propose a novel technique to adapt a large VLM to automated driving scenes.

## 2 RELATED WORK

**Vision-language Models** Large pre-trained vision-language models have shown great potential in learning representations that are transferable across a wide range of downstream tasks in zero-shot manner. Radford et al. (2021) proposed an efficient way to learn image representations on large scale data by making use of contrastive pre-training on image-caption pairs. Zhai et al. (2022) analysed contrastive training to align image and text models and found that locked pre-trained image models with unlocked text models work best. A systematic study of scaling laws for contrastive learning is conducted by Cherti et al. (2023). They have identified power law scaling for multiple downstream tasks. The connection between text and other modalities has received less attention. The closest approach to our work is LidarCLIP (Hess et al. (2023)), which learns a mapping from lidar point clouds to a pre-existing CLIP embedding space, effectively relating text and lidar data with the image domain. Our work was inspired by that idea, however, we consider a new input modality, i. e. radar spectra. Girdhar et al. (2023) proposed a similar approach to train encoders of several modalities. In this paper we leverage vision-language models to achieve a better representation for radar spectra input.

**VLMs for Automotive Scene Understanding** There is an ongoing surge of research into VLM applications in the automotive domain, taking advantage of large language models' reasoning ability coupled with perception using foundational vision models. Recent surveys are provided by Zhou et al. (2023); Yang et al. (2023b). Automotive scene understanding with VLMs takes different forms: object detection (Najibi et al., 2023; Ding et al., 2023) and VQA (Qian et al., 2023; Dewangan et al., 2023) produce "narrow" descriptions, capturing a subset of scene elements, while we desire a full scene description. Captioning approaches (Ding et al., 2023) require expensive ground truth annotation, not available for our data. Our method is closest to Hess et al. (2023) and Romero et al. (2023) which take the retrieval approach, by matching the input scene measurement to an embedding vector which lies in a representation space shared with a language encoder - and can thus be queried using free text. However, the latter utilizes out-of-the-box CLIP, and is thus limited to its performance, while the former relies on a large-scale automotive dataset. By contrast, radar spectra data is limited and as we show performance of off-the-shelf VLMs is not satisfactory. Further, there were no prior attempts to train a radar-spectra model.

**Prompt Generation** Open vocabulary models can serve as classifiers across any arbitrary set of categories specified with natural language, called *prompts*, consisting of a set of hand-written templates (e. g., a photo of a {}, where {} is replaced by the class name). Zhou et al. (2022) showed that performance of such models can be sensitive to the prompt wording, proposing context optimization to learn continuous vectors as task-specific prompts. Mokady et al. (2021) leverage a pre-trained language model to obtain better understanding of both visual and textual data and can efficiently generate meaningful captions for large-scale and diverse datasets. Another framework, called BLIP by Li et al. (2022), addresses vision-language understanding and generation tasks and effectively utilizes noisy web data by bootstrapping the captions. For generating meaningful captions for the training of our proposed RSLM we leverage the above mentioned techniques.

**Object Detection on Spectra** Object detection using automotive radar spectra is attracting increasing interest since recent introduction of public datasets (Rebut et al. (2022); Wang et al. (2021); Paek et al. (2022)). Zhang et al. (2021) collected a radar dataset and proposed a one-stage anchor-based detector that generates both 3D and 2D bounding boxes. Wang et al. (2021) created the CRUW dataset and proposed an object detection network working on range-azimuth radar spectra. FFT-RadNet (Rebut et al. (2022) eliminates the overhead of computing the range-azimuth-Doppler tensor, learning instead to recover angles from a range-Doppler spectrum. It is trained both to detect vehicles and to segment free driving space. Our proposed design choice for the spectra encoders and detection networks are based on FFT-RadNet. Decourt et al. (2022) presented DAROD, an adaptation of Faster R-CNN object detector for automotive radar on range-Doppler spectra. Giroux et al. (2023) proposed hierarchical Swin Vision transformers for radar object detection.

```
["there is a car 4 meters forward and a car 7 meters forth, this is
one vegetation in the scene, a building, and a pole",

"there are multiple objects, a rendering of the car, a fence, a
building, a traffic sign, a sidewalk, a wall, a road, and a sky",

"a cropped photo of at least 6 objects, there is a car 11 meters ahead
2 meters to the right, a car 22 meters in front less than 1 meter to
the left", ...]
```

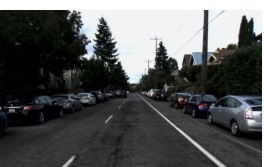

Figure 2: Example of created captions (left) for a frame (right) from CRUW dataset (Wang et al. (2021)) based on ground truth object positions and pseudo ground-truth classes. The pseudo ground-truth classes are obtained using a segmentation model.

## 3 PROPOSED APPROACH

Vision language models (Radford et al. (2021)) show promising performance across different tasks. We aim to harness this generalization ability to examine the semantic content of radar spectra measurements of automotive scenes. To address this, we train a radar spectra-language model, consisting of a spectra encoder and a text encoder with a shared embedding space, embedding vectors representing the observed scene. The latter is achieved by using the radar spectra-image measurement coupling in driving datasets to train the spectra encoder to match the embedding space of a VLM, following Hess et al. (2023). To obtain an embedding space that better fits our data we formulate a process for fine-tuning a VLM using generated captions of automotive scenes, without human annotations. We validate our approach in two ways: First, we evaluate our method on a spectra retrieval task using text queries, directly attempting to shed light on what elements of the scene are captured in radar spectra. Second, we inject the spectra embedding in a baseline spectra object-detection network to observe an improvement in detection performance.

The section is organized as follows: The datasets are presented in Section 3.1 and the proposed process of fine-tuning a VLM with automotive data is described in Section 3.2. We explain the training and architecture of the proposed radar spectra encoder in Section 3.3, and present its application for different down stream tasks in Section 3.4.

### 3.1 DATASETS

In this paper we use three datasets for autonomous driving: RADIal (Rebut et al. (2022)), CRUW (Wang et al. (2021)), and nuScenes (Caesar et al. (2020)). The RADIal dataset contains 8252 annotated frames. Each frame has a range-Doppler spectrum (512x256x16), lidar points, radar points, images, centers of cars, and free space annotations in birds-eye view. CRUW has 40734 annotated frames with range-azimuth spectra (128x128x8), images, centers of cars, pedestrians, and cyclists in range-azimuth coordinates. nuScenes includes lidar, radar point clouds, camera, IMU, and GPS data. In our work, we only use images from the train-validation split with 40157 samples.

### 3.2 CLIP FINE-TUNING

**Motivation** To create a radar spectra-language model we rely on embeddings from an image encoder of an existing VLM. However, publicly available VLMs are generally not specifically adapted to automotive scenes. Research by Radford et al. (2021) shows that the accuracy of zero-shot classification for KITTI dataset varies from 21% to 44%. This corresponds to our experience with this and other VLMs (see Table 1). Therefore, we fine-tune a baseline VLM for road scenes.

**Caption generation** In the context of CLIP fine-tuning, the requirement for image-caption pairs presents a challenge, particularly when dealing with datasets where only a limited number of object classes are represented. To address this limitation and expand the diversity of classes available for image-caption pairs, we adopt a two-step approach across all datasets. In the first step, we employ a segmentation model on each frame of the datasets, thereby augmenting the available classes by generating 19 classes. In the second step, we generate captions based on the augmented class set obtained in the previous stage and on ground truth information regarding the presence and positions of objects within the images. The caption generation process increases the variety of captions by omitting part of the present objects, permuting caption components, and utilizing synonymous scene modifiers. See Fig. 2 for an example. This way, multiple different captions are generated for each dataset frame, based on the objects which are present in this frame.

**Image Size Mismatch** CLIP image encoder typically accepts square images, obtained by resizing and center cropping the input. However, our datasets have a width-to-height ratio greater than 1.6, leading to a loss of information on the edges when cropped to a square. To mend this, we experiment with two alternative approaches: a) positional encoding interpolation, maintaining the aspect ratio, and b) generating three square crops aligned to the left, right and center. These crops are fed to image encoders producing three outputs, which are averaged to obtain one image embedding. For more details please refer to Appendix A.4.

**Losses** We utilize the following loss functions: the InfoNCE loss from Radford et al. (2021), which is a variant of contrastive loss wherein weights are inversely the co-occurrence of frame classes, and the focal loss proposed by Lin et al. (2017). A comprehensive description of these loss functions is available in Appendix A.4.

**Fine-tuning methods** We compare several fine-tuning techniques, spanning four distinct approaches. The first is fine-tuning the entire model, the second introduces fully-connected adapters on the outputs of the frozen model. The third method we explore is LoRA, low-rank adaptation of large language models Hu et al. (2021), a well-established fine-tuning technique for transformer-based models, which introduces learned corrections to pre-existing frozen weights. A recent study by Zhai et al. (2022) demonstrated that employing a robust pre-trained image encoder while exclusively fine-tuning the text encoder yields superior results compared to fine-tuning the entire network. Therefore, as the fourth method, we chose to fine-tune only the text encoder. Experimental evaluation of the methods is provided in Section 4.1.

## 3.3 RADAR ENCODER TRAINING

**Method** To obtain paired radar-spectra and text encoders we use a similar concept as presented in Hess et al. (2023). We train a radar encoder to output similar embeddings to a VLM image encoder. During training of the radar encoder, the VLM model is frozen (see Fig. 1a). Matching pairs of radar spectra and images are input to the network: an image to the frozen VLM image encoder and a corresponding radar spectrum to the radar encoder. We train the radar encoder to minimize the difference between the image and radar encoder outputs, where both outputs, the radar embeddings and image embeddings, have the same size. In this work, we compare two networks for the radar encoder: a network with CNN backbone and a network with a Feature Pyramid Network (FPN) backbone. The architecture of the RSLM is depicted in Fig. 1b. Additional details are provided in Appendix 5.

**CNN radar encoder** The CNN network includes three convolutional layers, batch-normalization, average pooling, a fully-connected layer and a layer normalization. The parameters of the first convolutional layer depend on the radar spectrum type and the number of input channels of the dataset at hand.

For the RADIal dataset, which includes range-Doppler spectra, we use the recommended paramters given by Rebut et al. (2022). For the CRUW dataset, which consists of range-azimuth spectra, the parameters have to be chosen differently. For more details on the parameters, see Section 4.2.

**FPN radar encoder** We choose the Feature Pyramid Network (FPN) of FFT-RadNet Rebut et al. (2022) as the radar encoder. Detection, and segmentation heads are not included in the radar encoder. A convolution layer and fully-connected layer are added, to project the output to the same space as the CLIP embeddings. The parameters of the first convolutional layer in the radar encoder are the same as for the CNN radar encoder, and depend on the dataset at hand.

## 3.4 DOWNSTREAM TASKS

To investigate the effect of the learned radar spectra feature embeddings for different downstream tasks, we consider object detection as well as free space space estimation as two applications. We combine the trained radar encoder with a detection segmentation network.

### 3.4.1 DETECTION AND SEGMENTATION NETWORKS

We choose FFT-RadNet as used by Rebut et al. (2022) as our detection backbone. The parameters of the first convolution layer in the radar encoder are set as described in Section 3.3. We use both the detection and the driveable space segmentation heads, as defined in Rebut et al. (2022). The loss

Table 1: **Left:** Average weighted accuracy on CRUW dataset for off-the-shelf VLMs. Models: CLIP Radford et al. (2021) ViT-B/32 and ViT-L/14 , EVA-CLIP Sun et al. (2023) ViT-L/14, BLIP Li et al. (2022) ViT-B/32 (base image-text matching model) and Open CLIP Cherti et al. (2023) ViT-L/14 pretrained on datacomp_xl. **Right:** Average weighted accuracy on CRUW dataset for fine-tuned models.

| label | CLIP ViT-B/32 | CLIP ViT-L/14 | EVA-CLIP | BLIP | Open CLIP | Basic | W/o weights | FC adapter | Focal loss | LoRA | Pos. emb. interpol. | Frozen image enc |
|---|---|---|---|---|---|---|---|---|---|---|---|---|
| sidewalk | **0.60** | 0.48 | 0.47 | 0.42 | 0.55 | 0.47 | 0.37 | **0.65** | 0.52 | 0.61 | 0.53 | 0.59 |
| fence | 0.53 | 0.53 | **0.63** | 0.52 | **0.63** | 0.49 | 0.55 | 0.51 | 0.57 | **0.60** | 0.53 | 0.59 |
| traffic light | 0.46 | 0.47 | 0.48 | 0.50 | **0.59** | 0.50 | 0.55 | 0.54 | 0.51 | **0.59** | 0.57 | 0.58 |
| traffic sign | **0.64** | 0.52 | 0.52 | 0.46 | 0.61 | 0.50 | 0.64 | 0.36 | 0.63 | 0.61 | 0.60 | **0.68** |
| person | 0.56 | 0.59 | 0.69 | 0.53 | **0.72** | **0.92** | 0.61 | 0.49 | 0.57 | 0.71 | 0.60 | 0.75 |
| rider | 0.45 | **0.57** | 0.54 | 0.56 | 0.57 | **0.98** | 0.56 | 0.56 | 0.52 | 0.54 | 0.53 | 0.59 |
| car | 0.44 | **0.59** | 0.50 | 0.49 | 0.56 | **0.70** | 0.62 | 0.49 | 0.61 | 0.58 | 0.60 | 0.54 |
| truck | 0.54 | **0.55** | 0.54 | 0.51 | 0.55 | **0.86** | 0.72 | 0.51 | 0.52 | 0.55 | 0.57 | 0.61 |
| bus | 0.42 | 0.49 | 0.45 | 0.45 | **0.58** | **0.81** | 0.76 | 0.40 | 0.46 | 0.47 | 0.45 | 0.52 |
| train | **0.72** | **0.72** | 0.57 | 0.52 | 0.57 | 0.73 | **0.80** | 0.53 | 0.65 | 0.53 | 0.63 | 0.56 |
| motorcycle | **0.57** | 0.52 | 0.46 | 0.51 | 0.46 | **0.96** | 0.85 | 0.44 | 0.48 | 0.48 | 0.45 | 0.44 |
| bicycle | 0.69 | 0.60 | 0.50 | 0.52 | **0.70** | 0.54 | 0.64 | 0.50 | 0.64 | **0.66** | 0.60 | 0.61 |
| Mean | 0.55 | 0.55 | 0.53 | 0.50 | **0.59** | **0.71** | 0.64 | 0.50 | 0.56 | 0.58 | 0.56 | 0.59 |

function is defined as $L = L_{\text{det}} + \lambda L_{\text{seg}}$, where $0 < \lambda \in \mathbb{R}$ a weighting factor and the detection and segmentation losses are defined as follows:

$$L_{\text{det}} = \text{focal}(y_{\text{class}}, \hat{y}_{\text{class}}) + \beta \text{ smooth-}L_1(y_{\text{reg}} - \hat{y}_{\text{reg}}),  \tag{1}$$

$$L_{\text{seg}} = \sum_{r,a} \text{BCE}(y_{\text{seg}}(r,a), \hat{y}_{\text{seg}}(r,a)),  \tag{2}$$

where $0 < \beta \in \mathbb{R}$; $\text{focal}(y_{\text{class}}, \hat{y}_{\text{class}})$ is the focal loss for true $y_{\text{class}}$ and predicted $\hat{y}_{\text{class}}$ class labels; smooth-$L_1$ is the smooth $L_1$ loss, where $y_{\text{reg}}$ and $\hat{y}_{\text{reg}}$ denote the ground-truth polar coordinates of the centers of the bounding boxes and the predicted ones, respectively. BCE denotes the binary cross entropy loss for true free-space map $y_{\text{seg}}$ and predicted map $\hat{y}_{\text{seg}}$, where $r$ and $a$ stand for range and azimuth coordinates, respectively.

### 3.4.2 DETECTION AND SEGMENTATION NETWORKS WITH RADAR EMBEDDINGS

We propose to inject embeddings from the pre-trained radar encoder (see Section 3.3) into the detection network. We hypothesize that this would introduce a semantic prior benefiting detection. An illustration of the proposed architecture is provided in Appendix A.8. The input radar spectra tensor is concurrently fed into the detection backbone described above and the radar encoder. The radar feature embeddings output by the radar encoder, are transformed by an adapter branch to match the size of the output features of the detection backbone, and are summed with those. As radar encoder we use the FPN radar encoder. The loss function for training is as defined in Section 3.4.1.

## 4 EXPERIMENTS

In this section we examine the practical benefit of our proposed methods. In Section 4.1 we present experimental results for *fine-tuning the CLIP image encoder* for automotive scenarios. The semantic content of radar spectra is analyzed in Section 4.2, where the radar spectra-language model is evaluated on *classification* and *retrieval* tasks. Furthermore, in Section 4.3 the trained radar encoder is evaluated in combination with a detection head and a segmentation head on two downstream tasks: *radar object detection* and *free space space estimation*.

### 4.1 CLIP FINETUNING

**Evaluation of off-the-shelf VLMs** We compare multiple VLMs for use in our domain of interest, evaluating the zero-shot classification performance for 11 classes on the CRUW dataset, which are particularly relevant for autonomous driving, cf. Table 1. Following the methodology outlined in Radford et al. (2021), we forward the input image to the image encoder and provide two contradicting prompts to the text encoder, specifically ["there is a car", "there is no car"]. This process yields predicted probabilities for each text prompt given the image. Subsequently, we use these predicted probabilities to calculate the weighted accuracy $(TP/(TP + FN) + TN/(TN + FP))/2$ for each class.

Table 2: Comparison of top-1 and top-100 precision scores for retrieval task for original VLM, fine-tuned VLM and RSLM on CRUW dataset.

| | Top 10 | | | | Top 100 | | | |
|---|---|---|---|---|---|---|---|---|
| Label | Original VLM | Fine-tuned VLM | CNN RSLM | FPN RSLM | Original VLM | Fine-tuned VLM | CNN RSLM | FPN RSLM |
| sidewalk | 1 | 1 | 1 | 1 | 1 | 1 | 1 | 0.99 |
| building | 1 | 1 | 1 | 1 | 1 | 1 | 1 | 1 |
| wall | 0.9 | 1 | 0.4 | 0.9 | 0.8 | 1 | 0.72 | 0.86 |
| fence | 1 | 1 | 0.9 | 1 | 0.99 | 1 | 0.91 | 0.96 |
| traffic light | 0.3 | 0.2 | 0.1 | 0.1 | 0.57 | 0.05 | 0.05 | 0.06 |
| traffic sign | 1 | 1 | 1 | 0.8 | 1 | 1 | 0.88 | 0.87 |
| person | 0.2 | 1 | 0 | 1 | 0.68 | 1 | 0.08 | 0.95 |
| rider | 0.8 | 1 | 0.3 | 0.6 | 0.94 | 1 | 0.13 | 0.4 |
| car | 0.7 | 1 | 1 | 1 | 0.84 | 0.92 | 1 | 0.93 |
| truck | 1 | 1 | 0.6 | 0.7 | 1 | 0.85 | 0.28 | 0.25 |
| bicycle | 1 | 1 | 0.7 | 0.4 | 1 | 0.26 | 0.26 | 0.2 |
| Mean | 0.809 | 0.927 | 0.636 | 0.773 | 0.893 | 0.904 | 0.574 | 0.679 |

By taking into account the class dis-balance in the test datasets, this metric provides a more precise evaluation of the model performance than accuracy only. Due to the sensitivity of CLIP models to prompts, we build several sets of prompts for each class and compute the average over the weighted accuracy across all prompt sets. The results of this evaluation are presented in the left part of Table 1. For the retrieval task, the top-10 and top-100 retrieval metrics are located in Appendix A.3. Open CLIP demonstrated the highest performance, and thus, we chose to use this model in subsequent stages of our research.

**Datasets for fine-tuning** For fine-tuning we utilize datasets with road scenes: RADIal Rebut et al. (2022), CRUW Wang et al. (2021), nuScenes Caesar et al. (2020). For RADIal and CRUW, both images and ground truth labels are used. From nuScenes, only images are taken.

**Fine-tuning VLMs** We fine-tune the Open CLIP model using image-caption pairs, following the procedure outlined in Section 3.2. The setup parameters can be found in Appendix A.5. The following fine-tuned models are considered:

**Basic** The entire original Open CLIP model is fine-tuned using a weighted contrastive loss, followed by averaging the left, center, and right crop image embeddings.

**W/o weights** Basic model with unweighted contrastive loss.

**FC Adapter** Building upon the frozen basic model, the fully connected (FC) Adapter model incorporates two FC layers where $in\_features = out\_features = emb\_dim$. Only the adapter is fine-tuned.

**Focal Loss** In this variant of the basic model, the contrastive loss is replaced with a focal loss.

**LoRA** Low-rank adaptation of large language models is used for fine-tuning the basic model.

**Pos. Emb. Interpol.** The center crop is removed from preprocessing and the positional embeddings are interpolated to match the required size within the basic model.

**Frozen Image Encoder** Only the text encoder is subject to training, while the image encoder remains frozen.

**Fine-tuning results** We evaluate the fine-tuned VLMs on zero-shot classification in the same way as the off-the-shelf VLMs. The results are summarized on the right side of Table 1, The basic fine-tuned model outperforms other models by mean average weighted accuracy on the CRUW dataset. Therefore, this model is chosen for training our RSLM. Example activation maps for initial and fine-tuned model are provided in Appendix A.7.

## 4.2 RADAR SPECTRA-LANGUAGE MODEL

**Setup** We train a radar spectra encoder by matching the embedding of the corresponding image produced by the image encoder of a frozen VLM. Separate encoders are trained for the range-Doppler spectra from RADIal dataset and range-azimuth spectra from CRUW dataset. We trained the CNN and FPN radar encoder with mean squared error (MSE) loss for matching the embeddings. For more detailed description of the parameter setup please refer to Appendix A.2.

**Evaluation of RSLM** The trained RSLM and the VLMs are evaluated on a retrieval task using the same prompts as for fine-tuning. The CLIP text encoder and our trained radar encoder are used to get the radar spectra-language model predictions. To compute the model predictions, the cosine similarity of the text and radar spectra embeddings is computed, cf. Fig. 1b. We rank the retrieved data samples by the cosine similarity values.

**Results** Retrieval performance is shown in Table 2. The FPN radar encoder outperforms the CNN radar encoder by most of the labels and by mean top-10 and top-100 accuracy. For person prompt, FPN achieves better results than the original VLM, due to fine-tuning. Thus, the RSLM can be use successfully applied to retrieval tasks. In Fig. 3 images are shown, which correspond to the retrieved spectra with maximal cosine similarity value for the given caption. It shows, that the RSLM can retrieve objects and scenes like parking lots and trucks, which were not presented in the ground truth. This shows, that radar spectra and language can be connected using the RSLM.

### 4.3 OBJECT DETECTION AND FREE-SPACE SEGMENTATION WITH RSLM RADAR EMBEDDINGS

For evaluating the performance of detection and segmentation networks with a pre-trained radar encoder from the RSLM, we propose to compare key metrics with baseline network without radar encoder, network with not-trained radar encoder and network without detection backbone.

**Models** We use the following models: "baseline" network is the FFT-RadNet Rebut et al. (2022), a detection and segmentation network, cf. Section 3.4.1. "frozen-enc" is the network from Section 3.4.2 with a frozen, pre-trained radar encoder. "fine-tuned enc" is the same network as "frozen-enc", however, the radar encoder is fine-tuned on the last 10 epochs . "only frozen enc" includes the pre-trained radar encoder, radar adapter, detection and segmentation head only. "only fine-tuned enc" is the same as "only frozen enc", but the radar encoder is fine-tuned on last 10 epochs. "from-scratch" is a random initialized network with the "frozen-enc" architecture.

**Metrics** We evaluate the models on the RADIal dataset and use the same metrics as in Rebut et al. (2022): For evaluating the object detection task the mean average precision (mAP) and mean average recall (mAR) are computed. For free space segmentation the intersection over union (IoU) is used.

**Results** Table 3 summarizes object detection performance (mAP, mAR, and $F_1$-score) and segmentation performance (IoU) for the models described above. FFT-RadNet Rebut et al. (2022) serves as baseline. We were not able to reproduce the results reported in the paper using the provided hyperparameters and therefore cite them as given, along with the results of our training (marked with ∗). We also cite results for the optimized FFT-RadNet of Yang et al. (2023a) (hyperparameters haven't been made public). Additionally, the results of T-FFTRadNet by Giroux et al. (2023) are presented, which uses a Swin Liu et al. (2021) backbone as opposed to FPN in FFT-RadNet. FFT-RadNet and T-FFTRadNet have roughly similar performance. Since code and exact parameters are not available for T-FFTRadNet, we have choosen FFT-RadNet as baseline. The columns of the table correspond to model features: "detect backbone" signifies the use of the detection backbone (MIMO pre-encoder and FPN Radar backbone) of FFT-RadNet. "Swin" indicates the substitution of the FPN encoder with a Swin Transformer. "radar enc" denotes the incorporation of the radar encoder from RSLM. "RSLM weights" indicates the usage of weights from the RSLM model for the radar encoder; otherwise, it is randomly initialized."Fine-tuned enc" signifies that the radar encoder was fine-tuned during detection training; otherwise it is frozen. Results in the table exhibit performance improvements when adding frozen radar embeddings into the model architecture. This

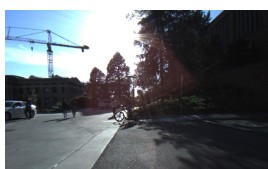

(a) a cyclist pedals through a radiant landscape

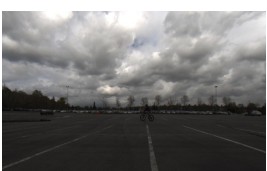

(b) Parking lot with many cars

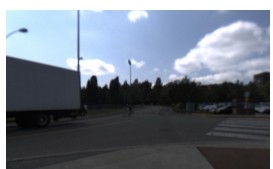

(c) truck cruising confidently on the open road

Figure 3: CRUW dataset images corresponding to spectra retrieved using the trained RSLM. The used query appears in the caption of each image.

Table 3: Ablation studies for detection (mAP, mAR, and $F_1$-score) and segmentation task (IoU). Top part lists baseline models, bottom lists different model architectures as described in Section 4.3, Models paragraph. In each result column, best results both for baselines and our models are in bold.

| Model | Detect backbone | Radar enc | RSLM weights | Fine-tuned enc | mAP (%) | mAR (%) | F1 (%) | IoU (%) |
|---|---|---|---|---|---|---|---|---|
| FFT-RadNet Rebut et al. (2022) | + | - | - | - | **96.8** | 82.1 | 88.8 | 74.0 |
| FFT-RadNet-optimized Yang et al. (2023a) | + | - | - | - | 92 | 89 | **90.4** | 76.7 |
| T-FFTRadNet Giroux et al. (2023) | Swin | - | - | - | 89.6 | **89.5** | 89.5 | **80.2** |
| FFT-RadNet $^{(*)}$ (baseline) | + | - | - | - | $88.8 \pm 1.7$ | $81.2 \pm 1.8$ | 84.2 | $67.3 \pm 1$ |
| frozen enc | + | + | + | - | $90.7 \pm 1.1$ | $81.8 \pm 2$ | **86.0** | $71.2 \pm 2.3$ |
| fine-tuned enc | + | + | + | + | $90.4 \pm 1.2$ | $81.4 \pm 2.1$ | 85.6 | $69.9 \pm 2.6$ |
| only-frozen enc | - | + | + | - | $0.1 \pm 0$ | $2.4 \pm 0.6$ | 0.1 | $55 \pm 16.7$ |
| only fine-tuned enc | - | + | + | + | $0.0 \pm 0$ | $2.7 \pm 1.1$ | 0 | $59.1 \pm 9.9$ |
| from-scratch | + | + | - | + | $88.1 \pm 2.8$ | $82.9 \pm 0.7$ | 85.4 | $72.6 \pm 1.9$ |

$(*)$ baseline FFT-RadNet architecture from Rebut et al. (2022), trained by us.

enhancement is observed in both detection and free-space segmentation tasks, as compared to the baseline model without embeddings ("frozen enc" and "fine-tuned enc" vs. "baseline"). Fine-tuning the radar encoder does not improve object detection or segmentation performance. Furthermore, the model "from-scratch" with the same architecture as the "frozen-enc" variant, exhibits slightly higher IoU scores for free-space segmentation and similar detection performance compared to the "frozen-enc" model. In contrast, models that exclusively incorporate the radar encoder component of RSLM ("only-frozen enc", "only fine-tuned enc"), whether frozen or trained during the last 10 epochs, do not successfully accomplish the detection task.

This shows, that using the pre-trained radar encoder from RSLM improves performance in downstream tasks ("frozen enc" vs. "baseline"), i.e. learning the feature embeddings is helpful. Note, that the pre-training does not use any labeled radar spectra data, and the weights of the radar encoder lead to similar performance as weights trained in a fully supervised manner ("frozen enc" vs. "from-scratch"). We emphasize that improvements are achieved with the same hyperparameters as the baseline model by adding the RSLM radar encoder. Visualizations for detection and segmentation results can be found in Appendix A.9.

**Discussion**   The proposed RSLM relies on pre-trained vision-language models, therefore it depends on the quality of the captions. More captions would help to fine-tune the corresponding language model, yielding a better RSLM. This performance dependence on VLM is an obvious limitation of the RSLM, as the VLM has a limited performance on some rare cases (*e.g.*, it cannot recognize traffic signs).

The investigations show that the proposed RSLM can learn relevant features for scene retrieval, see Fig. 7. In this paper only scene-level descriptions have been used, however, object-level descriptions would be also beneficial in future work. Experiments show that the learned features are relevant for downstream-tasks. For those tasks state-of-the-art performance can be obtained without the need for any additional data, only by making use of image-radar pairs.

Radar measurements are not significantly affected by bad weather conditions or time of day, which is one main advantage. Therefore, while the proposed RSLM was trained on images that are taken at daytime, it can be expected to work as well in "rainy", and "night" scenarios. However, this is not possible to verify since images in difficult conditions are not available. Therefore, new publicly released datasets with such difficult weather conditions might be beneficial. As another future application, the proposed model might be used for radar data generation.

## 5   CONCLUSION

We developed a radar spectra-language model (RSLM), to the best of our knowledge the first such model, for automotive scenes. Our method makes use of vision-language models (VLMs), which we first fine-tuned on automotive image data to improve their performance. We investigate the semantic content of radar spectra, by querying the RSLM with text descriptions and evaluating radar scene retrieval. In this way the model can even be used to query for different object types, for which no corresponding labels exist in the dataset. Moreover, the proposed methods overcomes the scarcity of labeled radar spectra data, since no labeled radar dataset is needed to train the RSLM. Finally, we showed that the performance in downstream tasks can be improved by injecting radar feature embeddings from the RSLM into a detection and segmentation model.

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

# A APPENDIX

## A.1 RADAR SIGNAL PROCESSING CHAIN

An overview of the classical radar signal processing chain in provided in Fig. 4.

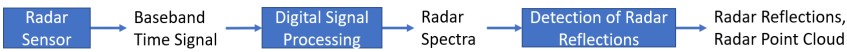

Figure 4: Overview of the classical signal processing chain of automotive radar sensors.

## A.2 PARAMETERS OF THE RADAR SPECTRA-LANGUAGE MODEL

RADIal range-Doppler spectra have been collected by $N_{T_x} = 12$ transmitters. Number of input channels for both CNN and FPN models equals two times the number of receivers $2N_{R_x} = 2 \cdot 16$, with real and imaginary signal components stored in different channels. The parameters are as follows: $kernel\_size = (1, N_{T_x})$, $dilation = (1, \delta)$, where $\delta = 16$ is the number of Doppler bins corresponding to the Doppler shift $\Delta$. $\delta = \frac{\Delta B_D}{D_{max}}$, with $B_D$ being the number of discretization bins for Doppler, and $D_{max}$ the largest Doppler value that can be measured. We use $kernel\_size = 3$, $dilation = 1$, $in\_channels = 8$ for range-azimuth spectra of the CRUW dataset. The full scheme of Radar spectra-language model can be found in Fig. 5.

We trained the CNN and FPN radar encoder with mean squared error (MSE) loss for matching the embeddings, Adam optimizer, learning rate $lr = 1e^{-3}$, and decay of 0.9 for every 10 epochs. Radar encoders with CNN backbone are trained on 30 epochs by default, the FPN radar encoders were trained on 45 epochs.

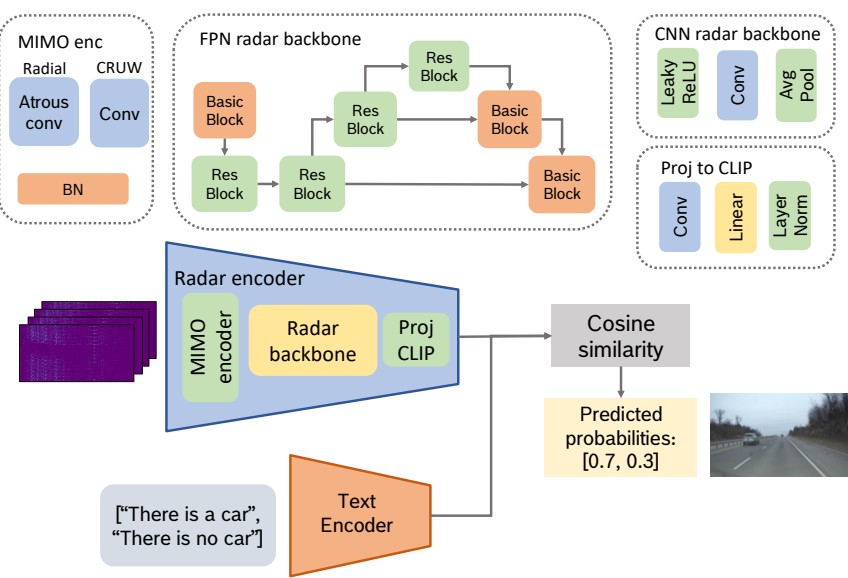

Figure 5: Radar spectra - language model.RSLM

## A.3 RETRIEVAL OF CLIP MODELS

The retrieval results for various CLIP models are detailed in Table 4. In particular, the Open CLIP and EVA models demonstrate identical top-10 mean precision scores. However, while EVA achieves a higher top-100 mean score, it registers a score of zero for the "car" category, rendering it unsuitable for our specific consideration. On the other hand, the CLIP ViT L/14 model is competitive with Open CLIP in terms of performance. However, taking into account higher weighted accuracy score (as shown in Table 1), we opt for the Open CLIP model.

```
# I_e[batch_size, emb_size]- image embeddings
# T_e[batch_size, emb_size] -  text embeddings
# t - learned temperature coefficient
# w[batch_size, ] - weights for captions in batch
# scaled pairwise cosine similarities [n, n]
logits = np.dot(I_e, T_e.T) * np.exp(t)
labels = np.arange(n)
loss_i = cross_entropy_loss(logits, labels, axis=0, w)
loss_t = cross_entropy_loss(logits, labels, axis=1, w)

# Contrastive loss
contrastive_loss = (loss_i + loss_t)/2

# Focal loss
p_image = softmax(logits, axis=0) # image to texts
p_text = softmax(logits, axis=1) # text to images
d_image = diagonal(p_image)
d_text = diagonal(p_text)
focal_loss = -(1 - d_text))**gamma log(d_text + eps)
            -(1 - d_image)**gamma log(d_image + eps)
```

$$L_{\text{contr}} = \frac{1}{2}(CE(e^t x \cdot y, \text{labels})$$
$$+ CE(e^t y \cdot x, \text{labels})) \qquad (3)$$

$$p_{\text{image}} = \text{softmax}(e^t x \cdot y)$$
$$p_{\text{text}} = \text{softmax}(e^t y \cdot x)$$
$$d_{\text{image}} = \text{diag}(p_{\text{image}})$$
$$d_{\text{text}} = \text{diag}(p_{\text{text}})$$

$$L_{\text{focal}} = - (1 - d_{\text{image}})^\gamma \log(d_{\text{image}} + \epsilon)$$
$$- (1 - d_{\text{text}})^\gamma \log(d_{\text{text}} + \epsilon) \qquad (4)$$

where $CE$ is the cross-entropy loss function, $x$ the radar embeddings, $y$ the image embeddings, $t$ a trainable temperature parameter, and labels $= [0, 1, \ldots, \text{batch\_size} - 1]$.

(a)            (b)

Figure 6: Left: Numpy-like pseudocode for contrastive loss and focal loss computation. Right: Formulas for calculating contrastive loss and focal loss.

### A.4 LOSSES FOR CLIP FINE-TUNING

The contrastive loss which is used in CLIP Radford et al. (2021) maximizes the cosine similarity between embeddings of real pairs of image and text and minimizes it for incorrect pairs, see Eq. (3). The ideal batch configuration for contrastive loss entails a scenario where a single caption within a batch corresponds exclusively to one image within that same batch, and vise versa. Regrettably, in our specific case, the dataset exhibits a notable deficiency in diversity. Consequently, it is highly probable that a substantial number of captions is suitable to a single image and several images may align with a single caption. To address this issue and prevent the inclusion of similar images in a single batch, we have implemented a custom sampling strategy that excludes subsequent frames within a defined time window. In order to achieve this for captions, it is necessary to define a similarity between captions. Afterwards, we exclude samples with similar captions in a batch or change them to less similar samples, which is computationally costly. Since captions are randomly generated within each epoch, this computationally complex operation is repeated in each epoch. To reduce computational costs we propose to weight captions according to co-occurrence of classes across the entire datasets. Each sample is assigned one calculated weight, thus all generated captions of one sample have the same weight, simplifying the process for practicality, although it may not be entirely precise. We incorporate the weights in the cross-entropy functions during the computation of the contrastive loss to compensate for correlated captions. Another approach is to leverage the focal loss Lin et al. (2017), which compensates class imbalance. To utilize this loss in our context, we use the mean between the focal loss for image probabilities and the focal loss for text probabilities, see Eq. (4). Numpy-like code for both contrastive and focal loss is provided in the Fig. 6a.

### A.5 FINETUNING VLM SETUP

We fine-tune the Open CLIP model using image-caption pairs, following the procedure outlined in Section 3.2. The fine-tuning process uses the Adam optimizer with a learning rate of $1 \times 10^{-5}$, coupled with a step scheduler that adjusts the learning rate every 10 epochs, applying a decay factor of 0.9. Our models undergoes training for 100 epochs, employing a batch size of 32. The effective batch size is equal to the size of the entire dataset. The checkpoint with minimal loss on the validation set is chosen for evaluation.

### A.6 IMAGE SIZE MISMATCH

The CLIP image encoder expects squared images of size 224 or 336 pixels, which is the result of resizing and center cropping the source images. However, the width-to-height ratio - $r_{wh}$ in the considered datasets is greater than 1.6. Thus, cropping to a square results in loss of information on the left and right sides of the image. To solve this problem, we consider two approaches: a) positional encoding interpolation and b) average embedding over two or three square crops. In the first method a), we do not apply center crop, thus, we keep the aspect ratio of the image, i.e. height $= 224, \text{width} = 224 \cdot r_{wh}$, instead width $= 224$. To fit the image encoder to new input

Table 4: Comparison of top-1 and top-100 precision scores for retrieval task among various CLIP models on CRUW dataset.

| Label | Top 10 | | | | | Top 100 | | | | |
|---|---|---|---|---|---|---|---|---|---|---|
| | CLIP ViT B/32 | CLIP ViT L/14 | EVA | BLIP | Open CLIP | CLIP ViT B/32 | CLIP ViT L/14 | EVA | BLIP | Open CLIP |
| sidewalk | 1 | 1 | 1 | 1 | 1 | 1 | 1 | 1 | 1 | 1 |
| fence | 1 | 1 | 0.9 | 0.3 | 1 | 0.98 | 0.98 | 0.99 | 0.71 | 0.99 |
| pole | 1 | 1 | 1 | 1 | 1 | 1 | 1 | 1 | 1 | 1 |
| traffic light | 0.8 | 0 | 1 | 0.1 | 0.9 | 0.94 | 0.27 | 0.99 | 0.01 | 0.65 |
| traffic sign | 1 | 1 | 1 | 1 | 1 | 0.56 | 0.98 | 1 | 1 | 1 |
| person | 1 | 1 | 1 | 1 | 0.7 | 1 | 1 | 0.98 | 1 | 0.62 |
| rider | 1 | 1 | 1 | 1 | 1 | 1 | 1 | 1 | 1 | 0.96 |
| car | 0 | 1 | 0 | 1 | 1 | 0 | 0.91 | 0 | 0.93 | 0.99 |
| truck | 1 | 1 | 1 | 1 | 1 | 1 | 0.99 | 0.94 | 0.91 | 0.97 |
| bus | 0 | 0.4 | 0.7 | 0 | 0 | 0 | 0.17 | 0.55 | 0.01 | 0 |
| bicycle | 1 | 1 | 1 | 1 | 1 | 1 | 1 | 1 | 1 | 1 |
| Mean | 0.8 | 0.85 | 0.87 | 0.76 | 0.87 | 0.77 | 0.85 | 0.86 | 0.78 | 0.83 |

size, we interpolate positional encoding to a new rectangular size. In method b), we perform three crops: the left crop is the square aligned to the left side, right crop the square aligned to the right side and center crop. Since the height-to-width ratio in our datasets is less than two, the two crops (left and right) capture the full picture with an overlap. However, using three crops exhibits better performance compared to using two crops. After this, three feature outputs are computed using the image encoder. Subsequently, we take the average of the three feature embeddings as the final output of the image encoder.

## A.7 Visualizations of class activation maps

To provide visual interpretations of the neural network decisions, we employed the GradCam algorithm Selvaraju et al. (2017). It gathers activations and gradients on the target layer, multiplies them, and projects the magnitude of the positive signal onto the initial image. We applied this technique to both the initial OpenCLIP model (Fig. 7b) and our fine-tuned OpenCLIP model (Fig. 7c-d). Our best fine-tuned model utilizes the mean of three image embeddings corresponding to the left, center, and right crops. Consequently, we generated activation maps for each input crop. The prompt "there is a person" was used as input for the text encoder, the image depicted in Fig. 7 from CRUW dataset was used for the image encoder. Fig. 7b shows that the regions with highest activation are spread on different regions of the input image, but not on the relevant part, i. e. the person on the bike next to the truck. In contrast, for our fine-tuned OpenCLIP model in Fig. 7c-d, the regions with highest activation (left and right crop) or large activation (center crop) are located directly on the relevant person. This shows qualitatively, that our method focuses on the correct region, and fine-tuning the VLM for automotive scenes is helpful.

## A.8 Fusion radar embeddings with the detection and segmentation network

The scheme of fusing the radar embeddings with the detection and segmentation network is presented in Fig. 8. The picture of the detection backbone is taken from Rebut et al. (2022) and adapted.

## A.9 Visualization results of object detection and free space segmentation

**Setup** All networks are trained on the RADIal dataset for 70 epochs, with Adam optimizer, learning rate $lr = 1e^{-4}$, decay of 0.9 for every 10 epochs. $\lambda$ is set to 100 for computing the training loss in (Eq. (1)), and $\beta = 100$ in the detection loss (Eq. (1)). A checkpoint with the best mean average precision on the validation dataset is chosen for evaluation.

A detection example is illustrated in Fig. 9. We conducted a comparative analysis involving the ground truth bounding box, the bounding box generated by FFT-RadNet, and our network, which incorporates an additional radar encoder from RSLM. Both FFT-RadNet and our model provide predictions for the range and azimuth of targets. These predictions are then projected onto Cartesian coordinates, and bounding boxes are created with fixed dimensions (4 meters in length and 1.8 meters in width) for cars. In the left part of the Fig. 9, we display the resulting bounding boxes and

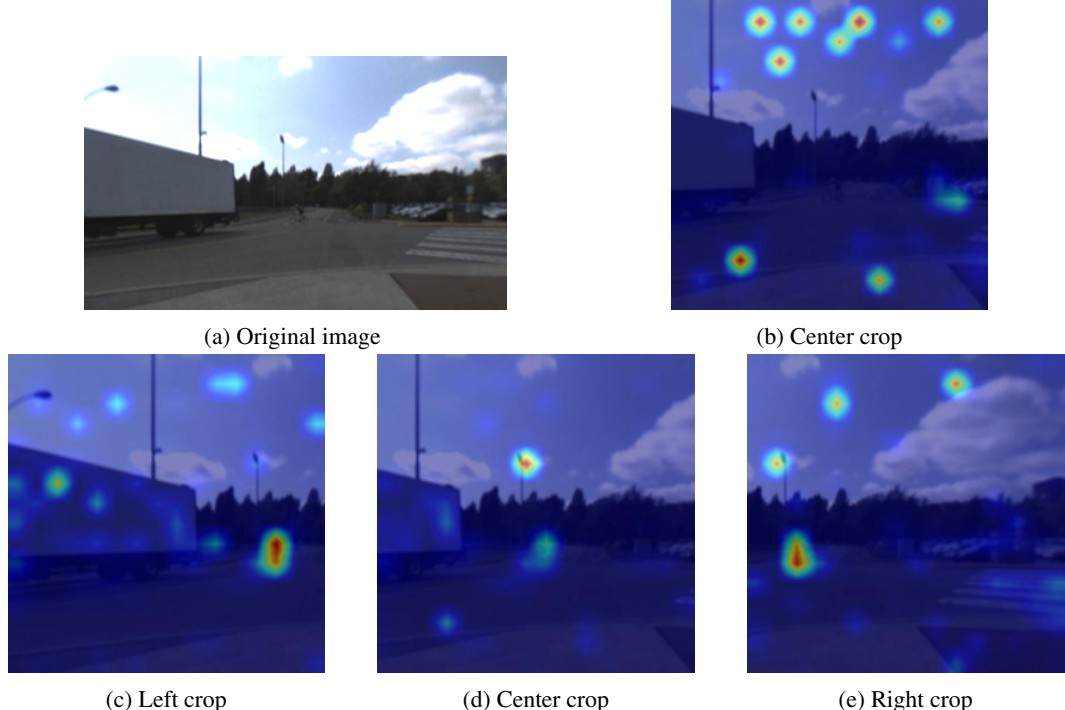

(a) Original image          (b) Center crop

(c) Left crop        (d) Center crop        (e) Right crop

Figure 7: **(b)** GradCam for OpenCLIP with the prompt: "there is a person". The areas with activation are not located on the relevant part of the image, i. e. the person next to the truck, showing that the model does not focus on the correct areas of the image. **(c-e)** GradCam for our fine-tuned OpenCLIP model with prompt: "there is a person" . The relevant region, the person next to truck, exhibits the highest activation values (left crop and right crop) or large activation values (center crop), showing that the model takes into account the relevant region.

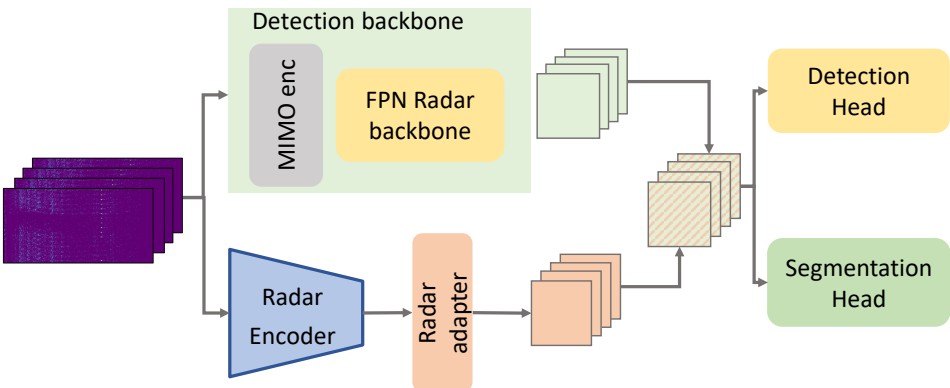

Figure 8: Fusion of detection network outputs and radar encoder from RSLM

augment them with lidar points for better visualization. To project the predictions onto the image, a 3D bounding box is constructed. We set the detection height to 0.5 meters and the car height to 1.5 meters. The right part of the Fig. 9 presents the projected bounding boxes within the image. Both visualizations show, that the prediction of our proposed model is better aligned to the ground truth bounding box, than the prediction of FFT-RadNet.

The illustration of free space segmentation is presented in Fig. 10. The red color signifies ground truth open driving space, green color represents free-space predicted by the corresponding model, and yellow color denotes an intersection of ground truth and predicted drivable space. Results for both FFT-RadNet and our model (labeled as "frozen_enc" in Table 3) are presented alongside the

ground truth for comparison with correspondign RGB image. Fig. 10 shows, that our proposed model predicts a free space region which aligns better to the ground truth than the one predicted by FFT-RadNet.

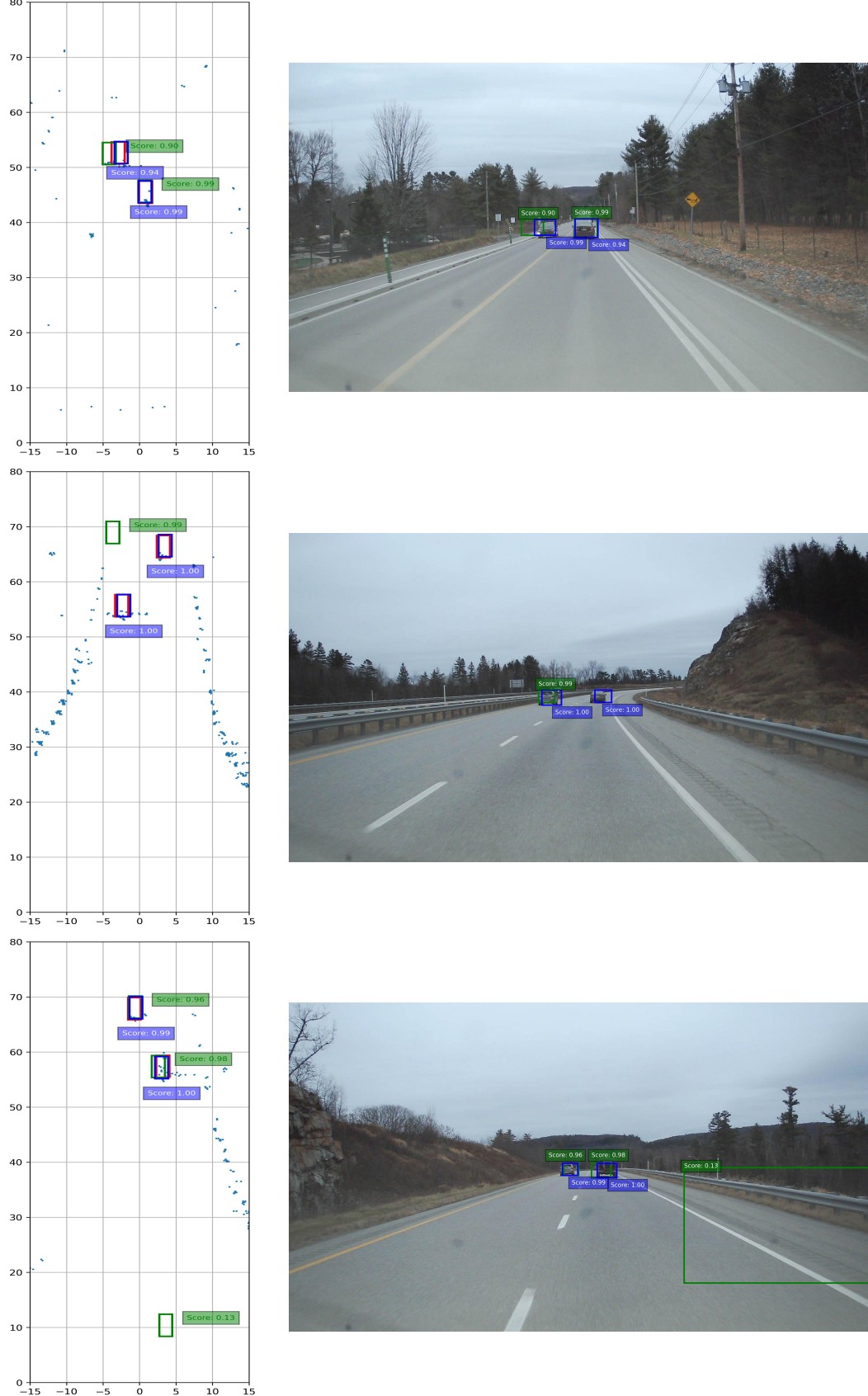

Figure 9: Detection results of FFT-RadNet (green) and our proposed network "frozen_enc"(blue). The bounding box prediction of FFT-RadNet is displaced w.r.t the ground truth bounding box (red), whereas the "frozen_enc" predicion aligns well with the ground truth. Confidence score equals 0.1. Left: Bounding boxes in Cartesian coordinates displayed on lidar point clouds. Right: Bounding boxes projected on image.

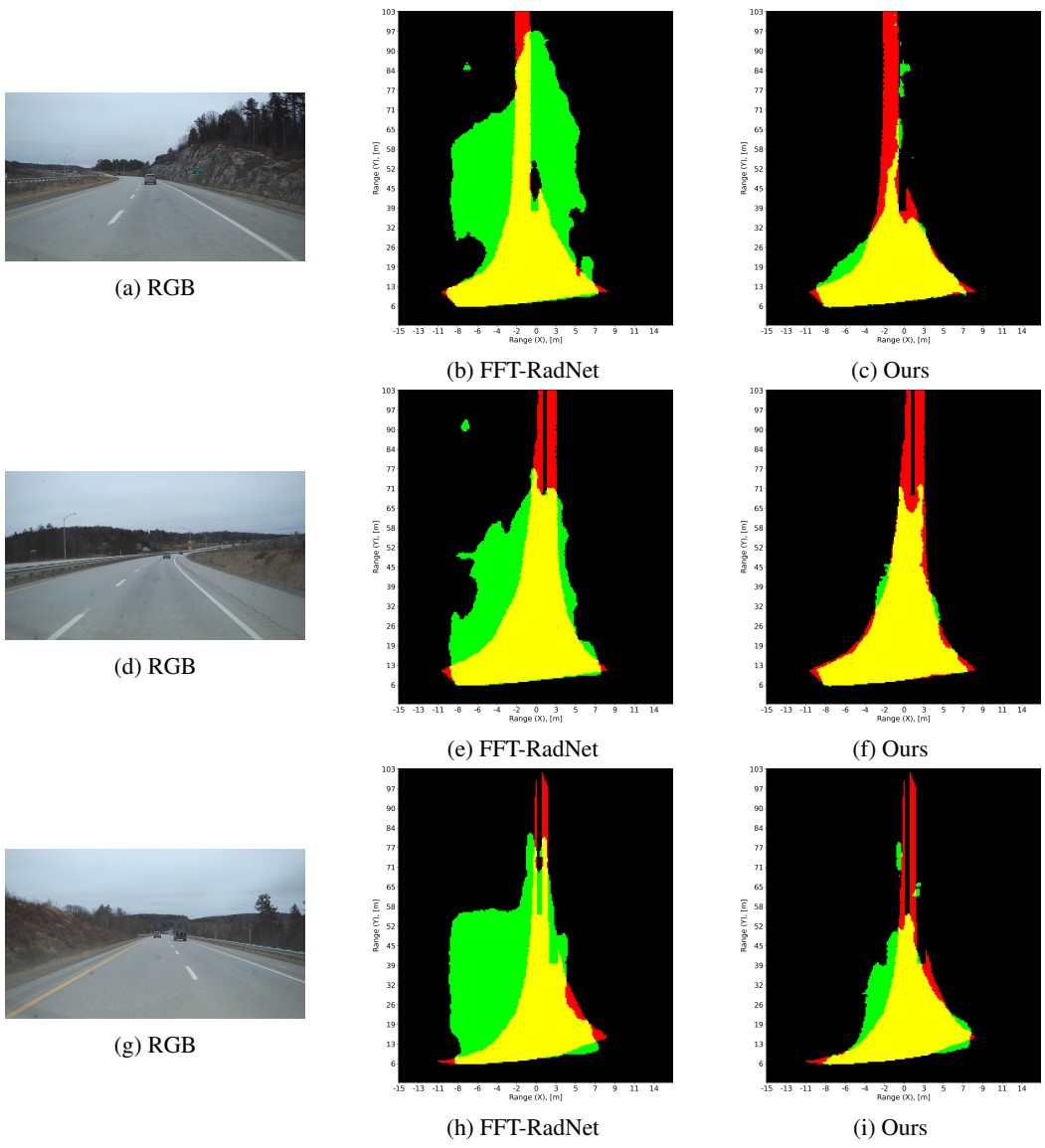

Figure 10: Example of segmentation result of FFT-RadNet and our network "frozen_enc". The predictions of our proposed method "frozen_enc" are better align to the ground truth. The red color signifies ground truth open driving space, green color represents free-space predicted by the corresponding model, and yellow color denotes an intersection of ground truth and predicted drivable space

