# OpenReview forum: "Radar Spectra-language Model for Automotive Scene Parsing"
_ICLR.cc/2024/Conference — Submitted to ICLR 2024_

### Official Review · Reviewer_Bffg · 2023-10-15

**Soundness:** 2 fair
**Presentation:** 2 fair
**Contribution:** 1 poor
**Rating:** 5
**Confidence:** 4

**Summary:**

This paper presents a study to investigate vision-language models for scene understanding in automotive scenes. To this end, a benchmark is created. For autonomous driving scene understanding, the benefits for downstream object detection and free space estimation are discussed. Ablation studies are well conducted and the paper is well structured.

**Strengths:**

1. The presented work is one of the first to train a radar-language model for autonomous driving scene understanding.
2. The paper is overall well structured.

**Weaknesses:**

1. Please consider comparing your proposed model against some existing adapted language models for driving scene understanding.
2. Please consider directly comparing your proposed model against some object detection and segmentation models to verify the superiority of your proposed model.
3. Most of the components from the proposed framework are from existing works. It is hard to find any novel technical designs in the presented framework. Please better clarify the technical novelty and theoretical contributions of the presented work.
4. Would you consider giving an overview of your proposed framework at the beginning of the methodology section, which can help the readers better understand the work?
5. The computation efficiency should be discussed, which is critical in automotive scene understanding.
6. The writing style and presentation quality could be further enhanced. In the introduction, the space between different paragraphs should be enlarged.

Sincerely,

**Questions:**

Would you consider presenting some visualization results of object detection and free space segmentation to qualitatively verify the effectiveness of your proposed method?

While there are not many works on radar-language models, there are extensive works on adapting large-language models for driving scene understanding. Would you consider discussing the relations and differences between your work and existing works in the related work section?

Sincerely,

---

> ### Author Response · Authors · 2023-11-21
>
> **Thank you for your comments and suggestions!** We address the issues raised below
>
> > R1:S1 The presented work is one of the first to train a radar-language model for autonomous driving scene understanding.
>
> It is the first to train a radar spectra – language model, and the first to make use of the shared language model embedding space to shed light on the semantic content of radar spectra.
>
> Radar spectra is a unique form of measurement – it captures the spatial reflection intensity of the electomagnetic signal and thus the geometric structure of the environment. But it also captures the reflection intensity along the Doppler axis, i.e. it is (radial) motion-aware, and the reflection intensity depends on the material properties, as well as geometry. It is clear that radar spectra carry richer information than the widely used sparse radar reflection points, but it is not quite known what kind of information can be extracted. The current work aims to provide a way to shed light on this.
>
> > R4:W1 Please consider comparing your proposed model against some existing adapted language models for driving scene understanding.
>
> A review of VLMs for automotive scene understanding was added to the related work section (edited part highlighted in blue). All methods are quite new - concurrent to our work. None of the reviewed methods is quite applicable to our setting of retrieval – where data is scarce and semantic ground truth is absent (and off-the-shelf VLM performance is unsatisfactory).
>
> > R4:W2  Please consider directly comparing your proposed model against some object detection and segmentation models to verify the superiority of your proposed model.
>
> The comparison of our proposed model to an existing radar object detection and segmentation model is included in Table 3. The baseline model in Table 3  is FFT-RadNet of Rebut et al. (2022), the SOTA for the RADIal dataset, we clarified this in the table and the corresponding section 4.3 (updated portions of the manuscript are highlighted in blue).
> Following this comment we added to the comparison SotA results from two additional papers on the same dataset.
>
> > R4:W3  Most of the components from the proposed framework are from existing works. It is hard to find any novel technical designs in the presented framework. Please better clarify the technical novelty and theoretical contributions of the presented work.
>
> Our paper introduces a novel task of extraction of semantic information from radar spectra – an otherwise generally non human - interpretable modality, and introduces an approach to tackle it. Both of the above are the main novelties of the paper. It is fortunate, that the encoder of an existing object detection and segmentation network (FFT-RadNet Rebut et al. (2022)) can be used as radar encoder of the radar spectra language model. This way, not another specific architecture has to be designed for one specific use case.
>
> In addition, we describe a novel process of generating captions for and fine-tuning an image-language model to extract general semantic information from AD scenes.
>
> Finally, to validate our radar-spectra model we modify an object detection model acting on spectra by  injecting the embedding vector from our spectra encoder, and demonstrate it improves SotA on the RADIal dataset.
>
> The goal of the work is to explore the semantic content of radar spectra, which is otherwise very difficult.
>
> In the updated paper version, we explain our contributinos in a clearer manner (highlighted by blue text color). We will try to clarify this further in the final version.
>
> > R4:W4 Would you consider giving an overview of your proposed framework at the beginning of the methodology section, which can help the readers better understand the work?
>
> Done. Overview provided at the begining of the "Proposed Approach" Section 3.
>
> > R4:W5 The computation efficiency should be discussed, which is critical in automotive scene understanding.
>
> We agree that computational efficiency is an important property for models running on an embedded device. Nevertheless, the main focus of the paper is to a) get a better understanding of the content of radar spectra and b) to investigate the benefits of the learnt features for downstream tasks. The focus is not on computational efficiency and the method is not intended to be run on an edge device, but rather is a research tool.
>
> In other words, the current work aims to provide a tool to examine the semantic content of radar spectra, rather than provide a perception algorithm for an autonomous vehicle. For this reason, computational efficiency is not a major consideration.

---

> > ### Author Response · Authors · 2023-11-21
> >
> > > R4:W6 The writing style and presentation quality could be further enhanced. In the introduction, the space between different paragraphs should be enlarged.
> >
> > Thank you for pointing this out. Issue with paragraph spacing has been corrected. We made numerous edits and clarifications (marked in blue) throughout the document which hopefully enhanced its presentation clarity.
> >
> > > R4:Q1 Would you consider presenting some visualization results of object detection and free space segmentation to qualitatively verify the effectiveness of your proposed method?
> >
> > We added a new appendix section A.6 with visualization of object detection and segmentation, and a brief discussion.
> >
> > > R4: Q2 While there are not many works on radar-language models, there are extensive works on adapting large-language models for driving scene understanding. Would you consider discussing the relations and differences between your work and existing works in the related work section?
> >
> > Done, added a review of VLMs for automotive scene understanding. Most works are very new, indeed concurrent to ours. None takes our approach, contrary to us relying on annotated data or using a frozen VLM. (And of course none does a radar spectra language model).

---

> > > ### Comment · Reviewer_Bffg · 2023-11-22
> > > **Comment**
> > >
> > > The reviewer would like to thank the authors for the added responses and clarifications, which helped solve some concerns.
> > >
> > > For this reason, the reviewer would like to elevate the rating accordingly.
> > >
> > > Sincerely,

---

> > > > ### Author Response · Authors · 2023-11-23
> > > >
> > > > We thank the reviewer again for the review!
> > > >
> > > > A new version was uploaded, with some further (hopefully) improvements to figures and to the presentation, and without highlighting.

---

### Official Review · Reviewer_9UPS · 2023-11-01

**Soundness:** 4 excellent
**Presentation:** 3 good
**Contribution:** 2 fair
**Rating:** 5
**Confidence:** 4

**Summary:**

In this paper, the authors propose a radar spectra-language model (RSLM). The RSLM is built upon CLIP with image as a bridge between radar and text. The RSLM is evaluated by a retrieval task and two downstream tasks. Experiments show that the RSLM has good zero-shot retrieval ability and can boost the performance of two downstream tasks.

**Strengths:**

1. To the best of my knowledge, this is the first paper trying to build a radar spectra-language model.
2. The fine-tuned VLM for autonomous driving scenes works much better than the off-the-shell CLIP.
3. The zero-shot retrieval ability of RSLM is impressive, especially for the small objects such as pedestrian and cyclist.

**Weaknesses:**

1. The author seems to lack paper writing skills. All the figures are unaesthetic bitmaps with low resolution and some of the figures are not necessary. For Figure 4a, it is better to use formulation instead of python code to describe the loss functions. For Figure 4b, such a simple architecture may be put in the supplement material.
2. Changing the position encoding without finetuning may cause performance drop, and splitting the image may break some objects on the edge. A better and more common way is to pad black pixels to the top and bottom of the image.
3. For the detection and segmentation downstream tasks, it is better to show some cases that the pretrained model helps improve the performance, not just numbers.
4. For autonomous driving tasks, the localization abilitiy is more important than the classification. Could you provide some visualizations such as attention map or GradCAM to see if the retrieved objects are corresponding to the right location?

Other issues:
1. For Equation 4, a period should be added in the end of the formula.
2. All the quotation marks are single quotes. Please use backquote in front of the quoted phrases.
3. Some of the RADIal is mistaken by RADiaL.

Overall, the proposed model is novel and the zero-shot results are interesting and impressive. However, the writing and lack of discussion lowered the final score. I would like to increase my score if the authors would polish the paper and redraw all the figures using vectorgraph.

**Questions:**

See above

---

> ### Author Response · Authors · 2023-11-21
>
> > R3:W1 The author seems to lack paper writing skills. All the figures are unaesthetic bitmaps with low resolution and some of the figures are not necessary. For Figure 4a, it is better to use formulation instead of python code to describe the loss functions. For Figure 4b, such a simple architecture may be put in the supplement material.
>
> Figures were re-exported in vector format.
>
> Figure 4 has been moved to the appendix and re-numbered to Figure 6. Formulation of loss functions was added in Section A.2.
>
> The reason we used python code was to be consistent with the CLIP paper Radford et al. 2021, which defined its InfoNCE loss with python code.
>
> > R3:W2 Changing the position encoding without finetuning may cause performance drop, and splitting the image may break some objects on the edge. A better and more common way is to pad black pixels to the top and bottom of the image.
>
> We always fine-tuned the model when using each of the adaptation methods we reported ( (1) positional encoding interpolation, (2) multiple square crops).
>
> When creating crops we made sure there's an overlap, hopefully mitigating effects like mentioned - of breaking an object on the edge, i.e. an object would usually appear in its entirety at least in one of the crops.
>
> Following this comment we tried using zero-padding, which worsened results (average weighted accuracy in the experiment did not exceed 0.5).
>
> We believe that the reason is that images in the RADIal dataset have aspect ratio of nearly 1:2, while the network input is 244x244 square. In this case (resize and) padding results in half the input image being black, with effective image height just 122 pixels, and many details are lost – resulting in the observed performance hit.
>
> > R3:W3 For the detection and segmentation downstream tasks, it is better to show some cases that the pretrained model helps improve the performance, not just numbers.
>
> We added a new appendix section A.6 with visualization of object detection and segmentation, and a brief discussion.
>
> > R3:W4 For autonomous driving tasks, the localization abilitiy is more important than the classification. Could you provide some visualizations such as attention map or GradCAM to see if the retrieved objects are corresponding to the right location?
>
> We added a new appendix section A.4 with visualization of VLM activations using GradCAM before and after fine-tunning.
>
> > Other issues:
> > - For Equation 4, a period should be added in the end of the formula.
> > - All the quotation marks are single quotes. Please use backquote in front of the quoted phrases.
> > - Some of the RADIal is mistaken by RADiaL.
>
> All done. Thanks for pointing out!

---

> > ### Comment · Reviewer_9UPS · 2023-11-23
> >
> > I appreciate the effort of authors. Most of my concerns are addressed except the presentation skills including figure drawing. I will slightly raise my score.

---

> > > ### Author Response · Authors · 2023-11-23
> > >
> > > We thank the reviewer for review and useful suggestions!
> > >
> > > We uploaded a final version with updated - hopefully, slightly improved - figures.

---

### Official Review · Reviewer_H2ZX · 2023-11-01

**Soundness:** 2 fair
**Presentation:** 3 good
**Contribution:** 3 good
**Rating:** 6
**Confidence:** 3

**Summary:**

The author proposes Radar-Spectra Language Model (RSLM) to help interpret the difficult modality (by humans).
In addition, the modality also does not have many datasets. Thus the approach is to leverage the expressive power Vision Language Model (VLM), and train a radar encoder to mimic the features produced by VLM.

RSLM first fine-tunes CLIP image encoder to road scenes (from self-driving car research).
The best image encoder for the task is OpenCLIP.
To connect radar spectra to the resulting CLIP features, RSLM trains a radar encoder to output features that are as similar as possible to the CLIP features.
The best radar encoder network is Feature Pyramid Network (FPN), and it is trained using MSE loss on retrieval tasks.
The resulting features then are inputted to a network that is trained on two downstream tasks: object detection and free-space estimation.
The object detection losses are focal and smooth-L1 loss, while free-space estimation is trained using BCE loss.

RSLM is tested to find the optimal components, e.g. usage of OpenCLIP, FPN.
In addition, it also analyzes the performance of RSLM on object detection and free-space estimation.
RSLM is able to surpass the baseline, FFT-RadNet, these two tasks.

**Strengths:**

The radar spectrum pre-training to optimize on similarity to fine-tuned OpenCLIP is novel. It allows for pre-training without a need for explicit Radar-spectra dataset.

**Weaknesses:**

No discussion on what is still hard to do or not reliable.
Also analysis of the varying the difficulty of the input scenes would help answer the previous question.

**Questions:**

What self-driving related take would Radar-spectra be able to do well, while other modality cannot or struggle with?

Typos:
Pg. 3, "Prompt Generation" section: (e.g. a photo of a {}) -- {} symbol should be replaced.

---

> ### Author Response · Authors · 2023-11-21
>
> **Thank you for the valuable feedback!** We would like to answer your questions:
>
> > R2:W1 No discussion on what is still hard to do or not reliable. Also analysis of the varying the difficulty of the input scenes would help answer the previous question.
>
> A discussion section was incorporated (page 10). The changes in the updated paper version are highlighted with blue text color.
>
> > R2:Q1 What self-driving related take would Radar-spectra be able to do well, while other modality cannot or struggle with?
>
> Radar has several advantages compared to other modalities, e.g., it is robust to difficult weather conditions like rain or sun glare, and it can directly measure the relative radial velocity of other objects. Moreover it can detect objects at large distances, depending on the sensor for example up to about 350m.  Nevertheless, the focus of the paper is not a comparison of radar to other modalities. We investigate the semantic content of radar spectra, which is otherwise difficult to interpret, unlike other modalities.
>
> > R2:Q2 Typos: Pg. 3, "Prompt Generation" section: (e.g. a photo of a {}) -- {} symbol should be replaced.
>
> Done! Thank you for pointing this out!

---

> > ### Comment · Reviewer_H2ZX · 2023-11-22
> >
> > Dear authors,
> >
> > Thank you so much for the replies. After reading them, and other discussions, I would like to keep my ratings as is.

---

> > > ### Author Response · Authors · 2023-11-23
> > >
> > > Thank you again for the review!
> > >
> > > We uploaded a final version, with some further (hopefully) improvements to figures and to the presentation, and without highlighting.

---

### Official Review · Reviewer_RqMG · 2023-11-01

**Soundness:** 2 fair
**Presentation:** 2 fair
**Contribution:** 2 fair
**Rating:** 3
**Confidence:** 4

**Summary:**

Radar sensors are integral to driver assistance systems and the future of autonomous driving due to their cost-effectiveness, long-range capabilities, and resilience to adverse weather. Typically, radar data is processed in point cloud format, but raw radar spectra contain more detailed information, though they are harder to interpret. This research focuses on enhancing radar spectra interpretability in the automotive context. It introduces a radar spectra-language model that enables natural language queries about scene elements within radar spectra. To address data scarcity, the study aligns the embedding space of a vision-language model. By fine-tuning for automotive scenes, it improves performance. This learned representation benefits scene parsing, enhancing free space segmentation and object detection when integrated into a baseline model.

**Strengths:**

This paper introduces the text information into feature fusion for radar spectra interpretability.

**Weaknesses:**

1.	The framework seems to be a simple combination of existing methods. I didn’t see the specific design for the radar spectra language model.
2.	The experiment of detection is not compared with SOTA methods such as RODNet.
3.	What is [20] in Table 3?
4.	If the description includes multiple object information, how do you align the text information with the corresponding object?

**Questions:**

If the description is not accurate will the information mislead the model?

**Details Of Ethics Concerns:**

No concerns.

---

> ### Author Response · Authors · 2023-11-21
>
> **We thank you for your valuable comments and suggestions!** Below we address the issues raised.
>
> > R1:W1 The framework seems to be a simple combination of existing methods. I didn’t see the specific design for the radar spectra language model.
>
> Our paper introduces a novel task of extraction of semantic information from radar spectra – an otherwise generally non human - interpretable modality, and introduces an approach to tackle it. Both of the above are the main novelties of the paper.
>
> In addition, we describe a novel process of generating captions for and fine-tuning an image-language model to extract general semantic information from AD scenes.
>
> Finally, to validate our radar-spectra model we modify an object detection model acting on spectra by  injecting the embedding vector from our spectra encoder, and demonstrate it improves SotA on the RADIal dataset.
>
> > R1:W2 The experiment of detection is not compared with SOTA methods such as RODNet.
>
> and
>
> > R1 :W3 What is [20] in Table 3?
>
> Thank you for pointing this out! The reference in table 3 is the FFT-RadNet model from Rebut et al. (2022) - it has been fixed in the updated paper version. FFT-RadNet is the SOTA model for the RADIal dataset, to which we compare our proposed methods. We consider the RADIal dataset for the evaluation of the proposed radar spectra language model. Unfortunatelly, RODNet cannot be applied to this dataset directly, since RODNet is designed to work on a sequence of complex valued range-azimuth-spectra, the data provided in the CRUW dataset. In contrast, the RADIal dataset contains range-Doppler spectra, where the azimuth dimension has not been processed yet.
>
> Following this comment we've added results of two more SotA methods for RADIal to the comparison. We've extended Sec. 4.3, in particular the ``Results" paragraph to accomodate for the changes. Modified portions of the manuscript are highlighted in blue.
>
> > R1:W4 If the description includes multiple object information, how do you align the text information with the corresponding object?
>
> In this work, we formulate queries as descriptions on scene level, we do not align text information with a specific object explicitly. When generating captions, we have the entire information of the scene. In conclusion – the alignment problem does not arise, at least for the setting described in the paper.
>
> > R1:Q1 If the description is not accurate will the information mislead the model?
>
> In general, yes -  imprecise descriptions may have a negative effect of training. For example, using inaccurate descriptions for highway scenes, e.g. "there are 3 cars in the street" (as opposed to "on the highway"), decreased the quality of the scene classification. That's why we don't use GPT augmentations of captions, as it sometimes adds information that is not relevant to the image like "it's a sunny day, car goes down the street" for a picture with cloudy weather. Meanwhile, we added augmentations that are supposed not to violate caption-image aligning, e.g. "it's a photo of the environment with a car" and "there is a car".

---

> > ### Comment · Reviewer_RqMG · 2023-11-22
> >
> > Thank you for your effort in the rebuttal. My major concern is that I dont trust your text formation since your CLIP finetuning is based on random caption generation. You also answered my concerns "In general, yes - imprecise descriptions may have a negative effect of training.". Therefore, I decided to keep my ratings.

---

> > > ### Author Response · Authors · 2023-11-22
> > >
> > > Thank you for your reply.
> > >
> > > Just to be sure we would like to clarify that the generated captions are not completely random: The captions generated for a frame are based on the semantic segmentation of that frame, so the captions include the actual contents of the frame (portions of the captions are randomly dropped for the sake of data augmentation - this is why the captions are "random", but the remainder still reflects the contents of the particular frame).
> > >
> > > The caption for a frame might be inaccurate only if its semantic segmentation is erroneous (in terms of frame contents, not in terms of object boundaries) - however, note that even noisy captions might not mislead the model, especially if the dataset is big and diverse enough: large caption data scrapped from the internet that large VLMs are trained on is also noisy. Nevertheless, VLMs show impressive performance. One reason might be that soft cost function / training process (contrastive learning) can deal with noisy labels.
> > >
> > > In our answer to Q1 above we purely hypothetically refer to systematic errors in the captions that effect a large portion of the data - for example, replacing "highway" by "street" in say 30% of the cases. In contrast, the process we use for generating captions generally produces nearly perfect results on the level of scene description, i.e. it rarely misses completely scene content or makes up scene content in practice.
> > >
> > > The process of caption generation is described in section 3.2 and Figure 4 contains examples for generated captions. They may not describe the entirety of the scene contents, but are generally correct, and therefore helpful for contrastive training.
> > >
> > > We adapted the caption generation explanation in section 3.2 to make it clearer and reflect the above - we uploaded an updated version, changes marked in blue.

---

> > > > ### Author Response · Authors · 2023-11-23
> > > >
> > > > Thank you again for the review!
> > > >
> > > > We uploaded a new version, with some further (hopefully) improvements to figures and to the presentation, and without highlighting.

---

### Author Response · Authors · 2023-11-21

**We thank all the reviewers for providing valuable feedback!**

We greatly appreciate that you find the paper well structured [R4], you emphasize that it is "the first paper trying to build a radar spectra-language model" (RSLM) [R3] and that RSLM has impressive zero-shot retrieval ability [R3]. We thank you for pointing out our contribution of a novel proposed radar-spectrum pre-training [R2] and that the paper introduces text information for spectra interpretability [R1].

We will update our paper with your proposed suggestions and also include new visualization results.

Below you will find our answers to your main criticism and suggestions in detail.

Corrected portions of the updated manuscript are highlighted in blue.

---

### Meta-Review · Area_Chair_ugju · 2023-12-10

**Metareview:**

A radar-spectra language model is proposed where a radar encoder is trained to match the features produced by a CLIP variant which is fine-tuned on road scenes with captions augmented by a segmentation method. The key strength is exploration of radar modality, which is important in driving safety applications but where ability similar to VLMs is currently lacked. The key weaknesses are insufficient analysis and need for improved presentation.

Reviewer RqMG has potentially mis-understood the caption generation process, which is clarified by the author feedback as depending just on category names for the image segmentation outputs. But it is not clear which segmentation model is used for this, where the paper can discuss its robustness if a closed-set method is used and potential to hallucinate if open-vocabulary. The clarity of the paper can also be improved by describing the exact templates used for caption generation. But more broadly, the performance of the VLM is a limiting factor for the RSLM, so a better understanding of its limitations is justified. Reviewer Bffg raises the concern of limited technical contributions given that the framework relies on existing methods. Reviewers 9UPS recommends improving the presentation quality. Reviewer H2ZX notes an understanding is needed on what is hard or not reliable, or conversely where radar may present an advantage. Such an analysis is indeed important. The paper suggests that radar can be expected work well in adverse weather, which is true for the sensor, but the presented method also relies on VLMs for its effectiveness. So more extensive evaluations can be added to the paper, especially if it is possible to evaluate on conditions where radar is beneficial (such as adverse weather or low-light conditions), for which other datasets like K-Radar (NeurIPS 2022) might be relevant. Overall, the AC agrees with the majority opinion that the paper may not be accepted for ICLR. The authors are encouraged to resubmit to a future venue with improved presentation and analysis.

**Justification For Why Not Higher Score:**

While the need for a radar spectra language model is well-motivated, the impact is insufficiently demonstrated, especially in contrast to existing VLMs and over other modalities like RGB images or lidar.

**Justification For Why Not Lower Score:**

Not applicable.

---

### Decision · Program_Chairs · 2024-01-16

Reject